# Glucose-oxidase like catalytic mechanism of noble metal nanozymes

Jinxing Chen[1,2], Qian Ma[1,2], Minghua Li[1], Daiyong Chao[1], Liang Huang[1,2], Weiwei Wu[1,2], Youxing Fang [1✉] & Shaojun Dong [1,2✉]

Au nanoparticles (NPs) have been found to be excellent glucose oxidase mimics, while the catalytic processes have rarely been studied. Here, we reveal that the process of glucose oxidation catalyzed by Au NPs is as the same as that of natural glucose oxidase, namely, a two-step reaction including the dehydrogenation of glucose and the subsequent reduction of $O_2$ to $H_2O_2$ by two electrons. Pt, Pd, Ru, Rh, and Ir NPs can also catalyze the dehydrogenation of glucose, except that $O_2$ is preferably reduced to $H_2O$. By the electron transfer feature of noble metal NPs, we overcame the limitation that $H_2O_2$ must be produced in the traditional two-step glucose assay and realize the rapid colorimetric detections of glucose. Inspired by the electron transport pathway in the catalytic process of natural enzymes, noble metal NPs have also been found to mimic various enzymatic electron transfer reactions including cytochrome c, coenzymes as well as nitrobenzene reductions.

[1] State Key Laboratory of Electroanalytical Chemistry, Changchun Institute of Applied Chemistry, Chinese Academy of Sciences, Jilin, PR China. [2] University of Science and Technology of China, Hefei, Anhui, PR China. ✉email: fangyx@ciac.ac.cn; dongsj@ciac.ac.cn

Nanozymes—nanoparticles (NPs) that exhibit enzyme-like properties—overcome the drawbacks of natural enzymes, such as easy deactivation, high cost, and poor recyclability, and thus are highly promising alternatives to their natural counterparts with superiority[1–3]. At present, there are two key problems that existed in the research field of nanozymes, that is, low catalytic activity and poor specificity. To solve these problems, a strategy based on biomimetic simulation of the structure of natural enzymes, i.e., single-atom nanozyme ($FeN_5$) with super high oxidase catalytic activity, and a copper coordinated MOF-818 with high specificity for catechol oxidation, have been demonstrated successfully[4,5]. Although the biomimetic strategy has been proved to be very effective for the design of nanozymes, there are only a few nanomaterials with a similar structure to natural enzymes can be found.

In particular, Au NPs are structurally different from the active site of glucose oxidase (GOD) but show GOD-like activity, which is highly effective for the oxidation of glucose to produce $H_2O_2$ and has attracted substantial interest in biological detection and therapeutics[6,7]. Although GOD mimics are of high research significance, their development is very slow. At present, only a few works have reported nanomaterials other than Au present GOD-like properties[8,9]. It is noted that these sporadic reports seldom descript the catalytic processes. Given these limitations, the pursuit of understanding the catalytic mechanisms of Au NPs and other NPs has great significance in rationally designing efficient nanozymes with various and specific functions.

Generally, there are two approaches for substrate oxidation by natural enzymes: (1) oxygenation oxidation, in which $O_2$ serves as an oxidant and is incorporated into the reaction products, and (2) dehydrogenation oxidation, in which protons and electrons are removed from the substrate and transferred to an electron acceptor[10–12]. Oxygenation is quite common in various reactions, and active intermediates containing oxygen, i.e., superoxide anion radical ($O_2^{•−}$), hydrogen peroxide ($H_2O_2$), singlet oxygen ($^1O_2$), and hydroxyl radical (•OH), must be produced to oxidize the substrate. For the dehydrogenation reaction, the oxidant plays the role of an electron acceptor and does not react with the substrate directly[13]. Glucose oxidation in vivo occurs through the dehydrogenation reaction, which is catalyzed by dehydrogenase (i.e., glucose dehydrogenase (GDH)) with nicotinamide adenine dinucleotide or nicotinamide adenine dinucleotide phosphate NAD(P) as the electron acceptor or by oxidase (i.e., GOD) with flavin and $O_2$ as intermediate and terminal electron acceptors, respectively[14]. GDH and GOD catalyzed glucose oxidation share a lot of similarities, and a difference is that the reduced product of NAD(P) is stable under ambient conditions, while the reduced flavin catalyzed by GOD is sensitive to air and will further donate electrons to $O_2$ to generate $H_2O_2$. Understanding the catalytic mechanism of natural enzymes will be helpful to learn the GOD-like activity of Au NPs and further rationally design GOD mimics.

In this study, we utilized 2,2′-azino-bis(3-ethylbenzothiazoline-6-sulfonic acid) radical ($ABTS^{+•}$) instead of $O_2$ as an electron acceptor to study the GOD-like activity of Au NPs. In the presence of glucose, Au NPs can catalyze the reduction of $ABTS^{+•}$, demonstrating that Au NPs catalyzed the oxidation of glucose by direct dehydrogenation rather than active oxygen radicals. The dehydrogenation was further confirmed by electron paramagnetic resonance (EPR) spectroscopy, as the 5,5-dimethylpyrroline-N-oxide (DMPO)-H adduct was detected during the course of glucose oxidation. These results indicate that Au NPs catalyzes the transfer of electrons from glucose to $O_2$ and the reduction of $O_2$ to $H_2O_2$. Similar results were also found with various noble metal NPs (i.e., Pt, Pd, Ru, Rh, and Ir), except that the oxygen was reduced to $H_2O$ instead of $H_2O_2$. In addition, noble metal NPs can catalyze the reactions of a variety of biomolecules, which can be catalyzed by flavoenzymes in organisms. Therefore, we defined these NPs as coenzyme mimics.

## Results

**Catalytic activity: $O_2$ as terminal electron acceptor**. The noble metal NPs used in this study were synthesized in an aqueous solution with polyvinylpyrrolidone (PVP) as the stabilizer and $NaBH_4$ as the reductant. Transmission electron microscopy (TEM) results showed a high degree of NPs dispersion, with mean particle diameters of 5 nm for all samples (Figs. s1–s6).

We used the oxidation of glucose by $O_2$ as a model system to evaluate the catalytic activity of the Au NPs (Fig. 1a). The generated $H_2O_2$ was detected by horseradish peroxidase (HRP)-based colorimetric system using 3,3′,5,5′-tetramethylbenzidine (TMB) as a chromic indicator. After incubation of Au NPs with glucose for 10 min, the characteristic absorbance spectrum of $TMB_{ox}$ was observed when HRP and TMB were introduced into the supernatant (Figs. 1b and s7). However, the noble metal NPs of Pt–, Pd–, Ru–, Rh–, or Ir-catalyzed reactions did not produce detectable signals of $TMB_{ox}$. This result was consistent with previous researches, which reported that among the six noble metal NPs catalysts, only Au NPs possess GOD-like properties and can produce $H_2O_2$. It is worth noting that the absence of $H_2O_2$ in the product does not mean that the catalysts cannot catalyze the oxidation of glucose since $O_2$ can be reduced to $H_2O$ (4$e^-$ path) as well as to $H_2O_2$ (2$e^-$ path)[15].

**Catalytic activity: $ABTS^{+•}$ as terminal electron acceptor**. Natural GODs can transfer electrons not only from glucose to $O_2$ to generate $H_2O_2$ but also to other electron receptors, such as methylene blue ($MB^+$), ferrocene (Fc), and $ABTS^{+•}$[16]. Therefore, we utilized $ABTS^{+•}$ instead of $O_2$ as an electron acceptor to reveal the similarity in catalytic activities between GOD and noble metal NPs due to the high reversibility and obvious color change (Fig. s8)[17]. In the presence of glucose, the noble metal NPs catalyzed the reduction of $ABTS^{+•}$, leading to a decrease in absorption at 734 nm (Figs. 1c and s9), and the catalytic performance followed the order Au > Pt > Ru > Ir > Pd > Rh NPs (Fig. 1d). Since $ABTS^{+•}$ is a reversible electron receptor and does not directly react with glucose, the results suggest a two-step reaction in which glucose is oxidized first and then electrons are transferred to $ABTS^{+•}$ via the NPs[18,19].

In addition to the ability to mimic GOD, the NPs have oxidase-like and peroxidase-like properties; that is, the NPs use $O_2$ or $H_2O_2$ to catalyze the oxidation of chromogenic substrates such as TMB, ABTS, and o-phenylenediamine (OPD) (Fig. s10)[20]. The realization of oxidase and peroxidase properties is generally believed to require the production of reactive oxygen species to oxidize substrates[21]. To reveal the difference between glucose oxidation and TMB oxidation, we systematically compared the three enzymatic catalysis reactions. Au NPs cannot catalyze the oxidation of TMB in the presence of $O_2$. The other noble metal NPs had obvious catalytic properties, and the catalytic activity followed the order Pt > Ir > Pd > Rh > Ru > Au NPs (Figs. s11 and s12). Their oxidase-like activities were more pronounced under acidic conditions than under basic conditions and achieved optimal performances at pH 4. The peroxidase-mimicking catalytic activity followed the order Ir > Pt > Ru > Rh > Pd > Au NPs. The optimal pH was approximately 4.5 (Figs. s13–s15). According to the Nernst equation, the redox potentials increase with decreasing pH due to the participation of protons in the reaction[22]. Because $O_2$ and $H_2O_2$ are more active under acidic conditions than under basic conditions, the catalytic activities under acidic conditions are higher for these NPs. The oxidation

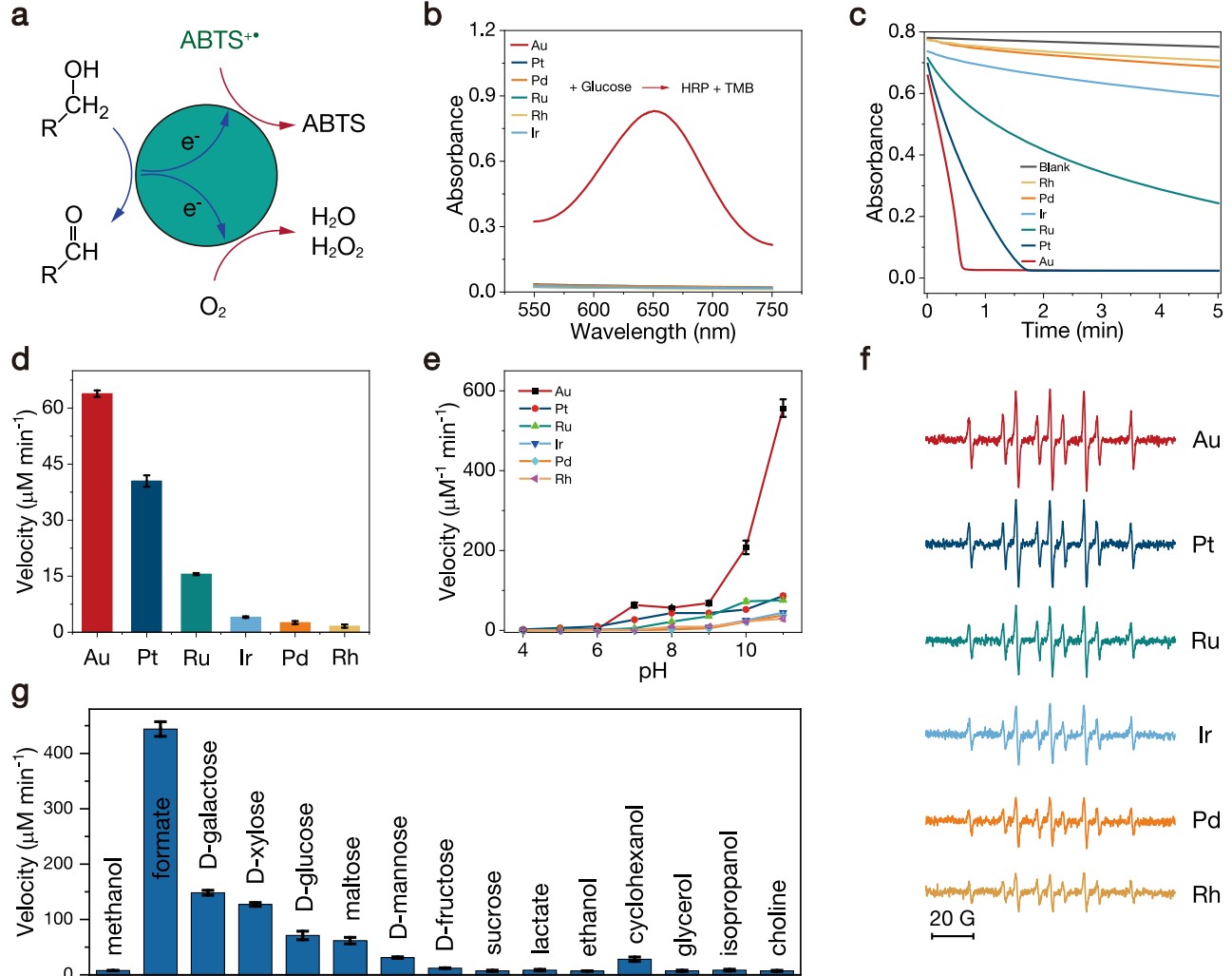

**Fig. 1 Glucose oxidation with $O_2$ or $ABTS^{+\bullet}$ as terminal electron acceptor. a** Catalytic oxidation of hydroxyl with $O_2$ or $ABTS^{+\bullet}$ as the oxidant. **b** Colorimetric detection of $H_2O_2$ produced by noble metal NPs catalyzing the aerobic oxidation of glucose. **c** Absorbance of $ABTS^{+\bullet}$ solution in the presence of glucose (50 mM) and different noble metal NPs (PBS, pH = 7.4). **d** Comparison of the reaction rates for $ABTS^{+\bullet}$ reduction. **e** pH-dependent reaction rates. **f** ESR spectra of the DMPO-H adduct from the DMPO and glucose reaction mixture in the presence of different noble metal NPs. **g** Au NPs-catalyzed $ABTS^{+\bullet}$ reduction in the presence of different substrates. Error bars represent standard deviation from three independent measurements.

potential of TMB is 1.13 V, which is just slightly lower than that of oxygen (1.23 V). Therefore, the oxidation of TMB requires the production of reactive oxygen species such as •O or •OH in the presence of $O_2$ or $H_2O_2$[23]. In contrast, the oxidation rate of glucose increased with increasing pH, implying that the oxidation of glucose does not rely on the oxidation ability of $O_2$.

**ESR tests.** The direct dehydrogenation of glucose catalyzed by noble metal NPs was further proven by electron spin resonance (ESR) spectroscopy with DMPO as the spin trapping agent (Fig. 1f). The reaction of Au NPs with glucose in degassed PBS in the presence of DMPO induced obvious ESR signals of DMPO-H adducts[24,25]. Since DMPO cannot dehydrogenate glucose, the reaction proceeds by abstracting hydrogen from glucose to form an Au–H intermediate and further transferring H to DMPO to generate DMPO-H adducts[26,27]. The other five noble metal NPs can also catalyze the dehydrogenation of glucose to form DMPO-H adducts, implying the same catalytic pathway applied.

Although most previous studies merely focused on the GOD-like properties of Au NPs, the catalytic capabilities of Au NPs are beyond simulating GOD. Essentially, the oxidation reaction of

glucose involves the dehydrogenation of the hydroxyl groups on glucose to form aldehyde groups. Therefore, we selected other hydroxyl-containing biomolecules to verify the ability of Au NPs to mimic other oxidases. Under the same reaction conditions, galactose was most easily oxidized among different sugars (Fig. 1g). We further tested some common molecules and realized general catalytic results, among which the oxidation rate of formic acid was the highest (Figs. s16 and s17). In a sense, we could define Au NPs as formate oxidase mimics.

**Electrode tests of $O_2$ reduction on NPs.** Natural GODs can transfer electrons not only from glucose to $O_2$ but also to electrodes, which conduced its wide usage in enzyme fuel cells[28]. This electron transfer pathway is also found in Au NPs (Fig. 2a). In the $N_2$-saturated electrolyte, after glucose was added, an obvious oxidation peak appeared, indicating that glucose was oxidized with the Au NPs (Fig. 2b). In the $O_2$-saturated electrolyte, the oxidation peak current decreased since the electrons collected by Au NPs from glucose can be transferred to oxygen. This phenomenon is also consistent with that of natural GOD. In the $O_2$-saturated electrolyte but without glucose, the reduction occurred

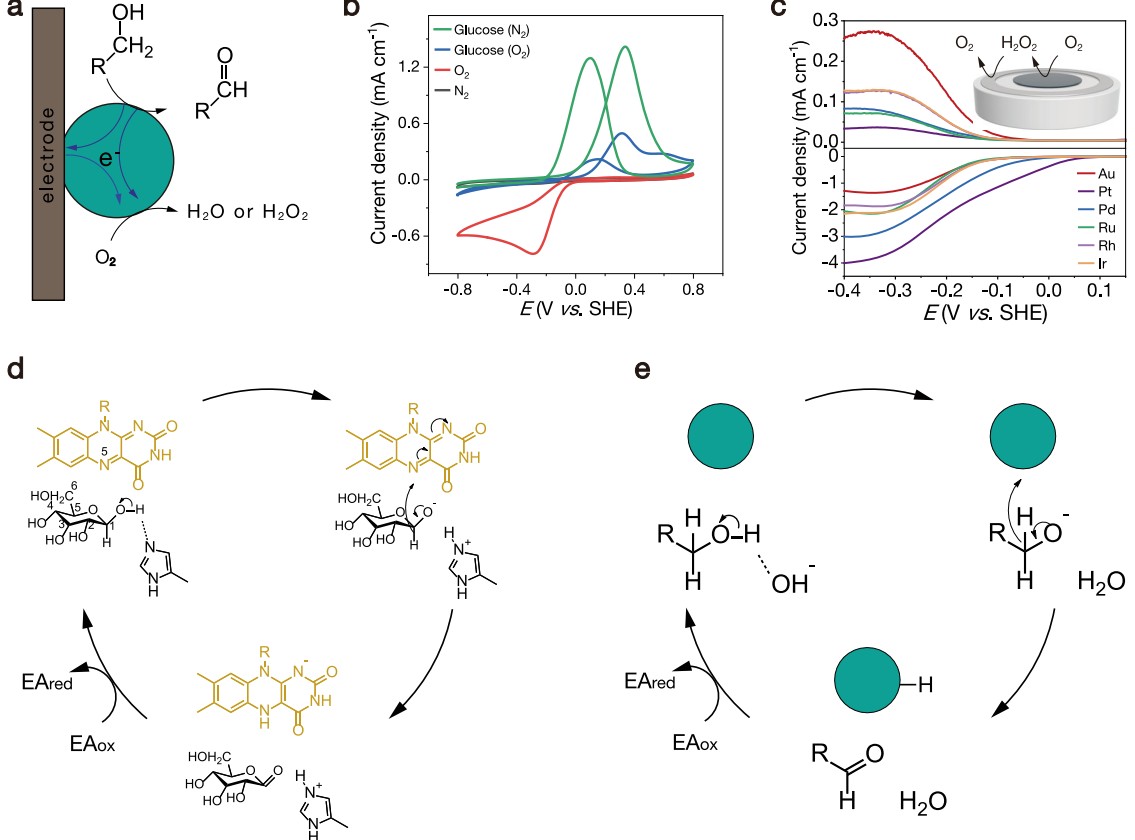

**Fig. 2 Mechanism of glucose oxidation catalyzed by noble metal NPs. a** Schematic illustration of electron transfer in the electrocatalytic oxidation of alcohols and the oxygen reduction reaction. **b** Cyclic voltammograms of Au NPs in $N_2$- and $O_2$- saturated 0.1 M NaOH solutions (with or without glucose). **c** RRDE measurements of the selective oxygen reduction of the catalysts in 0.1 M $O_2$-saturated NaOH electrolyte. **d** Mechanism of glucose oxidation catalyzed by GOD. **e** Mechanism of glucose oxidation catalyzed by noble metal NPs.

below 0 V in the negative scan, which proves that oxygen can acquire electrons from the Au NPs surface and conduct an oxygen reduction reaction (ORR). Because oxygen can be reduced to hydrogen peroxide and water by the two-electron and four-electron processes, respectively, we further used a rotating ring disk electrode (RRDE) to determine the electron transfer number of oxygen reduction on Au surface[29–31].

Au NPs were loaded on the glassy carbon disk electrode in the center (Fig. 2c). A potential of 1.2 V (versus the reversible hydrogen electrode) was applied on the ring (Pt) electrode. In a linear negative scan at a scan rate of 10 mV s$^{-1}$, $O_2$ was reduced by $4e^-$ to $H_2O$ or by fewer electrons to produce $H_2O_2$. $H_2O_2$ can be collected by the ring electrode and further oxidized to $O_2$. The amount of $H_2O_2$ was determined by the oxidation current at the ring electrode[32]. The electron transfer number of Au NPs for the ORR was calculated to be 2.38, and accordingly, the selectivity of $H_2O_2$ was 81% (Fig. s18). This result explains that $H_2O_2$ was produced during the glucose oxidation catalyzed by Au NPs. Notably, although the $H_2O_2$ selectivity of other noble metal NPs for the ORR was lower than that of Au NPs, $H_2O_2$ was also produced. However, no $H_2O_2$ was detected in the above Pt–, Pd–, Ru–, Rh–, and Ir NPs-catalyzed glucose aerobic oxidation experiments. It is speculated that these noble metal NPs can catalyze the decomposition of $H_2O_2$. We tested the catalase-like properties of these noble metal NPs, which catalyze the disproportionation of $H_2O_2$ to produce $O_2$ and $H_2O$ (Fig. s19). The absorption of $H_2O_2$ at 240 nm decrease with decomposition, which was utilized to test catalase activity by UV spectroscopy[20]. In the presence of a catalyst, especially Pt NPs, $H_2O_2$ decomposed

rapidly, and the catalytic activity followed the order Pt > Ru > Ir > Pd > Rh > Au NPs. Moreover, bubbles could be observed in the reaction vials, which visibly tracked that $H_2O_2$ decomposed to produce $O_2$. In addition, with an increasing pH, the decomposition rate of $H_2O_2$ increased for all noble metal NPs (Fig. s20). Among the catalysts, Au NPs did not show catalase-like activity. Together, these results give a reason that only Au NPs can catalyze the oxidation of glucose to produce $H_2O_2$, while other noble metal NPs cannot. The glucose oxidation catalyzed by Au NPs is fast, and the selectivity of $H_2O_2$ is high in the ORR step. Moreover, Au NPs do not catalyze the decomposition of $H_2O_2$. In contrast, the other noble metal NPs catalyze glucose oxidation relatively slowly with lower $H_2O_2$ selectivity than that of Au NPs, plus catalyzing the decomposition of $H_2O_2$.

**Mechanism of glucose oxidation**. The gathered results in our study draw a possible GOD-like mechanism for noble metal nanozyme-catalyzed glucose oxidation. For GOD, glucose is first activated by a close His residue (as a Brønsted base) to remove the C1 hydroxyl proton to form an intermediate (Fig. 2d)[33–35]. The removal of a hydroxyl proton from glucose facilitates hydride transfer from the C1 of glucose to the isoalloxazine ring of flavin adenine dinucleotide (FAD). Then, a direct hydride transfer occurs from the C1 position in glucose to the N5 position in FAD to produce FADH$^-$. FADH$^-$ is very sensitive to air and quickly oxidized by $O_2$ (electron acceptor, EA) to produce $H_2O_2$. The reaction path of Au NPs catalysis is the same as that of natural GOD, except that OH$^-$ is used as a Brønsted base to abstract H$^+$ from glucose (Figs. 2e and s21). The use of OH$^-$ as a Brønsted

base corresponds to that glucose oxidation and alcohol oxidation are faster under alkaline conditions[36]. In previous reports, theoretical calculation shows that the rate-determining step of alcohol oxidation is to break C–H bond, and Au has the strongest affinity on reducing the breaking energy of the C–H bond and thus exhibits the highest catalytic activity[37]. For Au catalyzed oxygen reduction, the energy barrier for breaking O=O double bond is higher, thus Au catalyzed oxygen reduction mainly takes the $2e^-$ path to produce $H_2O_2$[15]. Other noble metal NPs have the same catalytic process, except that $O_2$ is preferable to be reduced to water. In addition, both GOD and noble metal NPs can catalyze electron transfer from glucose to other electron acceptors (e.g., $ABTS^{+\bullet}$).

**Glucose detection**. Glucose detection has been developed by various methods and a normal one is based on the cascade reaction of natural enzymes (GOD and HRP). GOD catalyzes the oxidation of glucose to produce $H_2O_2$, and HRP conducts $H_2O_2$ to oxidize redox indicators such as ABTS, which indirectly indicates the concentration of glucose[38]. Accordingly, researchers used Au NPs instead of GOD and peroxidase mimics instead of HRP to realize the quantitative detection of glucose (Fig. 3a)[39–41]. Here we can use Au NPs as GOD mimics to oxidize glucose to produce $H_2O_2$, Prussian blue (PB) as the peroxidase mimic, and ABTS as an indicator to detect glucose. To produce considerable $H_2O_2$, the first step of glucose oxidation was carried out under alkaline conditions to achieve higher catalytic activity. Thereafter, the mixture solution was added to another acidic buffer solution. By using the peroxidase-like activity of PB, ABTS was oxidized by $H_2O_2$ to form $ABTS^{+\bullet}$ (Fig. 3c, left). The absorbance of $ABTS^{+\bullet}$ increased with increasing glucose concentration, which realized the glucose detection. The linear range of glucose detection was 0.1–0.5 mM, with a detection limit of 80 μM (Fig. 3d).

In view of the coenzyme-like properties of Au NPs, which can catalyze the transfer of electrons, we used $ABTS^{+\bullet}$ as both an electron acceptor and color indicator to detect glucose in one step (Fig. 3b). In the presence of glucose, the Au NPs could catalyze the reduction of $ABTS^{+\bullet}$, leading to a decrease in absorption at 734 nm (Fig. 3c, right). The absorbance of $ABTS^{+\bullet}$ decreased with increasing glucose concentration. The linear range of glucose detection was 0.01–2.5 mM, with a detection limit of 5 μM (Fig. 3d). Compared with the traditional two-step method based on GOD-HRP or GOD mimic-HRP mimic, this method is simple and rapid, with a wide detection range and low detection limit for glucose. Most importantly, this method breaks through the limitation that $H_2O_2$ must be produced in the two-step method. For example, although quite a few of NPs like Pt can catalyze the oxidation of glucose, the reaction hardly produces $H_2O_2$. Therefore, these NPs cannot be used to detect glucose by the two-step method. However, Pt NPs can directly use $ABTS^{+\bullet}$ as an electron acceptor to catalyze glucose oxidation and achieve glucose detection (Fig. 3d). The linear range of glucose detection was 0.02–5 mM, with a detection limit of 10 μM. This strategy dismisses $H_2O_2$ production in the process of glucose detection. It should be mentioned that these demonstrated model detections of glucose are still far from real sample detections that require excellent selectivity and stability of NPs, which may be overcome by the further rational modifications of NPs as well as designing cascade reactions with NPs.

**Coenzyme reduction**. Noble metal NPs can also transfer electrons from biomolecules to natural coenzymes such as flavin mononucleotide (FMN), coenzyme Q, and cytochrome $c$ (Cyt $c$) (Fig. 4a). Compared with the natural coenzyme, $O_2$ possesses a high reduction potential and is readily reduced as the electron acceptor. To obtain an oxygen-free environment, we chose formate as the reductant and Pt NPs as the catalyst to rapidly consume $O_2$ (Figure s22). The electron transfer from formic acid to FMN was monitored by UV–Vis spectroscopy. In the presence of Pt NPs and HCOOK, the absorption peaks of FMN at 450 nm

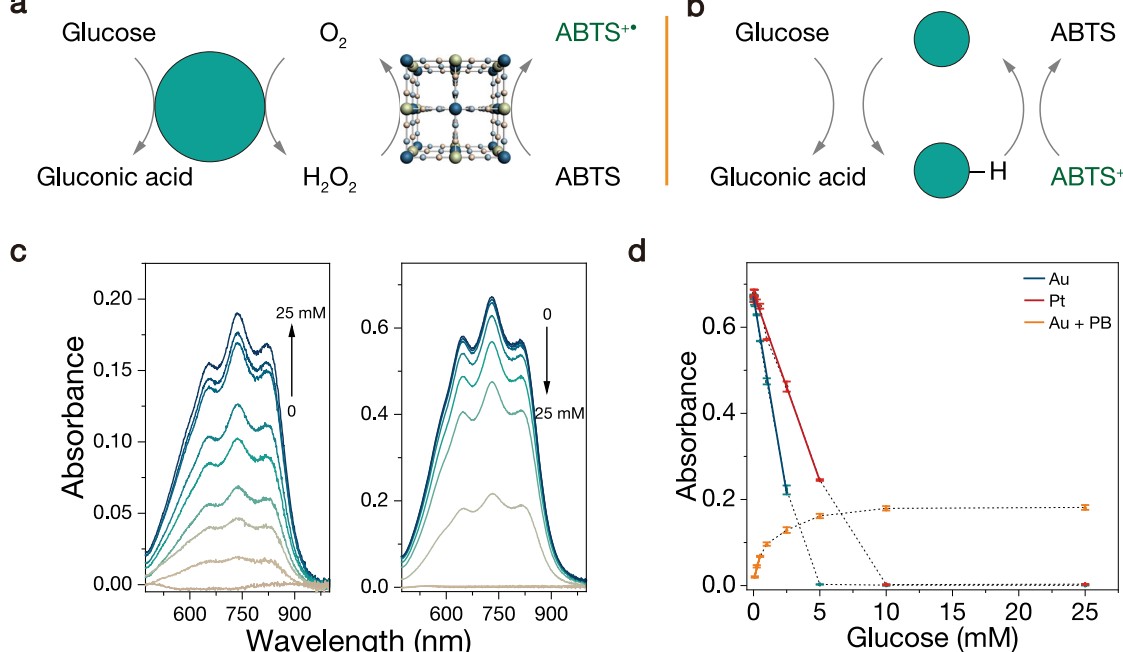

**Fig. 3 Two methods of glucose detection. a** Schematic illustration of glucose detection by the typical colorimetric assay with Au as a GOD mimic and Prussian blue as an HRP mimic. **b** $ABTS^{+\bullet}$ as an electron acceptor and colorimetric indicator for glucose detection. **c** ABTS oxidation catalyzed by Au and PB with $O_2$ and different concentrations of glucose (left). $ABTS^{+\bullet}$ reduction catalyzed by Au NPs with different concentrations of glucose (right). **d** Linear plot of absorbance intensity versus glucose concentration. All experiments were conducted in PBS (50 mM, pH = 7.4). Error bars represent standard deviation from three independent measurements.

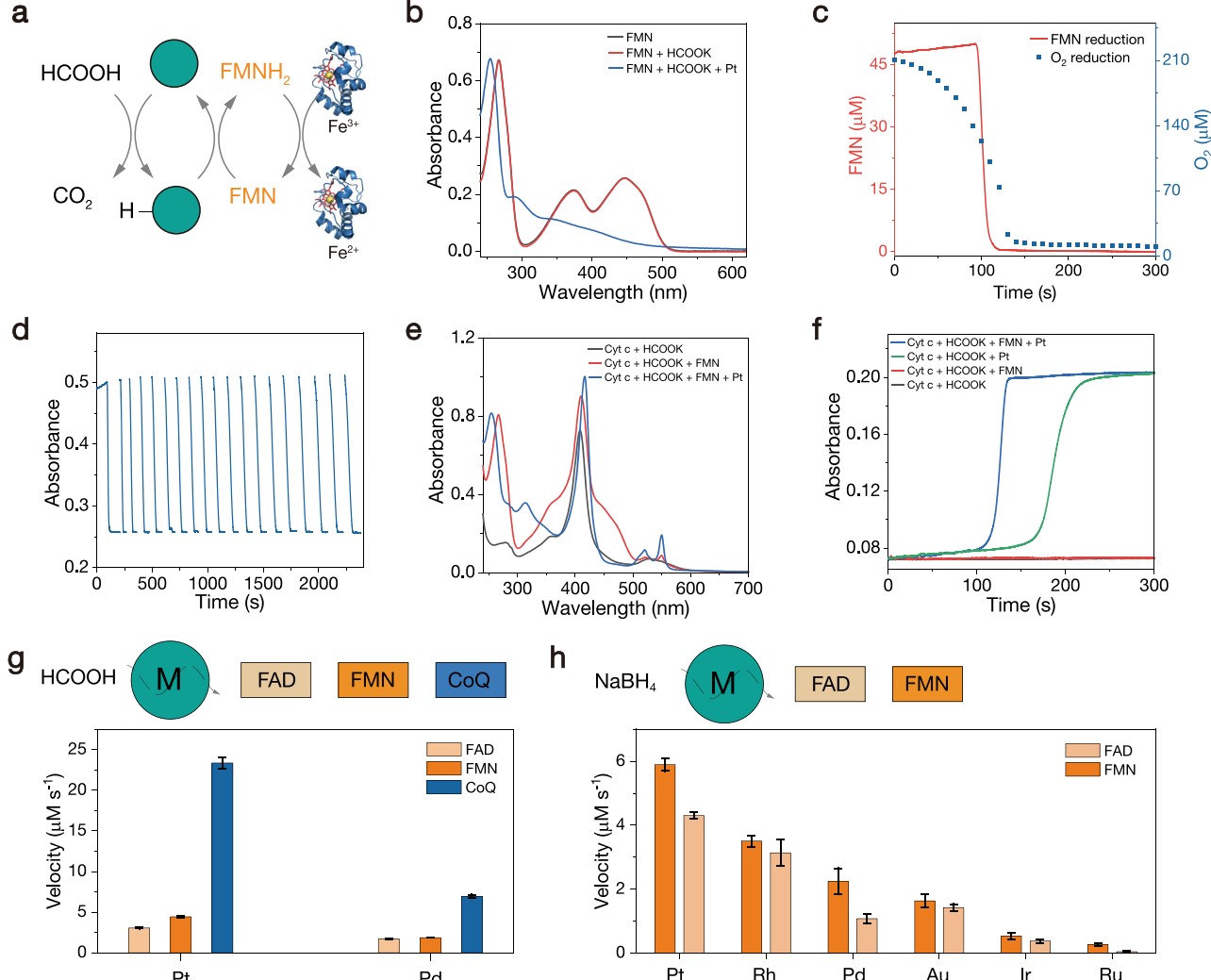

**Fig. 4 Coenzyme reduction. a** Schematic illustration of electron transfer from formic acid to FMN and Cyt *c* mediated by NPs. **b** UV–Vis absorption spectra of FMN, FMN + HCOOK, and FMN + HCOOK + Pt after reacting for 5 min. **c** The concentrations of FMN and $O_2$ versus time in the presence of HCOOK and Pt. **d** Repeated reduction and oxidation of FMN by HCOOK + Pt and air, respectively. **e** UV–Vis absorption spectra of different sample solutions after reacting for 5 min. **f** Kinetic curves of UV–Vis absorbance over time at 550 nm during Cyt *c* reduction under different conditions. **g** Catalytic performances of NPs for electron transfer from formic acid to coenzymes. **h** Catalytic performances of NPs for electron transfer from $NaBH_4$ to coenzymes. All experiments were conducted in PBS (50 mM, pH = 7.4). Error bars represent standard deviation from three independent measurements.

and 370 nm vanished, implying the reduction of FMN (Figs. 4b and s23)[42]. The control experiment did not exhibit any changes in absorption. This result demonstrated that Pt NPs can mediate electron transfer from formic acid to FMN to produce a reduced form of FMN ($FMNH_2$). Figure 4c shows the concentrations of FMN and $O_2$ over time in the presence of Pt NPs and HCOOK. The concentration of FMN maintained in the first 95 s and then rapidly decreased within ca. 10 s. The concentration of oxygen gradually decreased at the beginning of the reaction and was depleted in approximately 120 s. Since the reduction of $O_2$ is favorable over FMN, formic acid firstly consumed $O_2$, and the concentration of FMN remained unchanged because $FMNH_2$ is very sensitive to $O_2$ and easily reoxidized to FMN. Without $O_2$, FMN is reduced to $FMNH_2$, which is stable in the absence of $O_2$. The reduction of FMN catalyzed by Pt NPs and oxidation of $FMNH_2$ by $O_2$ were quite reversible (Fig. 4d). When shaking the reaction vessel, $FMNH_2$ was rapidly reoxidized to FMN due to refreshment of $O_2$, thereafter FMN has reduced again quickly when the reaction vessel was left standing.

$FMNH_2$ can further transfer electrons to ferric Cyt *c* to form ferrous Cyt *c*, which is a step in the electron transfer chain. In the presence of HCOOK, Pt NPs, and FMN, the Soret band peak of ferric Cyt *c* increased and redshifted from 409 to 414 nm and was accompanied by distinctly increased absorption of the α band (550 nm), indicating that ferric Cyt *c* converted to the ferrous form (Fig. 4e). In addition, electrons can be transferred directly from Pt NPs to ferric Cyt *c*. However, the reaction speed was slow, which may be caused by steric hindrance, considering that the heme iron is wrapped by a protein with a molecular weight of 12,000 Da in Cyt *c* (Fig. 4f). Pt NPs can catalyze the reduction of not only FMN but also FAD and coenzyme Q. Pd NPs also exhibited catalytic activities (Figs. 4g and s24–s27). For Au, Ru, Rh, and Ir NPs, $O_2$ was hard to exhaust. Therefore, it is difficult to use them to catalyze the reduction of coenzymes in the presence of formic acid. Alternatively, $NaBH_4$ can be used as a relatively strong electron donor to catalyze the reduction of coenzymes. Since $NaBH_4$ consumed $O_2$ rapidly, the catalytic activity of all noble metal NPs was observed (Fig. 4h).

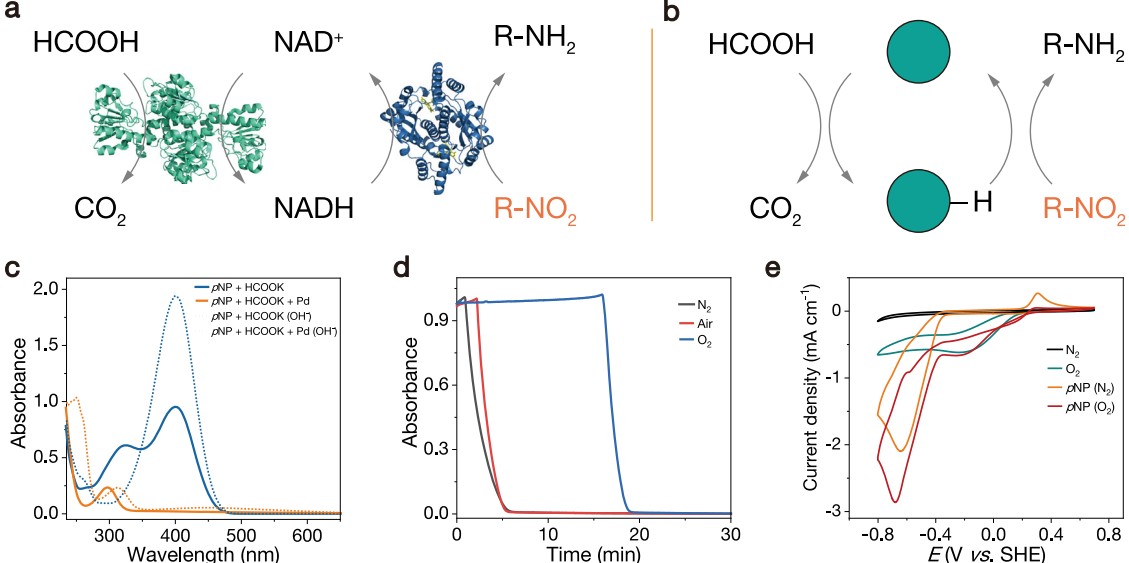

**Fig. 5 Nitro compound reduction. a** Schematic illustration of nitro reduction through the cascade reaction catalyzed by formate dehydrogenase coupled with nitroreductase. **b** Nitro compound reduction catalyzed by Pd NPs with formic acid as the electron donor. **c** UV–Vis absorption spectra of pNP (20 μM) in different mixture solutions after reacting for 10 min. Dotted line: NaOH was added before UV–Vis absorption measurement. **d** UV–Vis absorption of pNP (20 μM) at 400 nm versus time under different conditions. **e** Cyclic voltammograms of Pd NPs in the presence of pNP (10 mM), with a scan rate of 10 mV s$^{-1}$. All experiments were conducted in PBS (50 mM, pH = 7.4).

**Nitro compound reduction**. By learning the catalytic pathway of natural enzymes and the similarity of the catalytic pathway between nanomaterials and enzymes, we can flexibly apply nanomaterials to various catalytic reactions. For example, the reduction of nitro groups in nature is catalyzed by nitroreductase, with NAD(P)H as an electron donor. NAD(P)H is generated from the reduction of NAD(P) by dehydrogenases, such as formate dehydrogenase with formic acid as the electron donor (Fig. 5a)[43]. Due to the excellent catalytic activities of noble metal NPs for dehydrogenation and hydrogenation, we can directly use formic acid as an electron donor to reduce nitro groups (Fig. 5b). Among the six noble metal NPs, Pd NPs have the best catalytic performance, because of the rapid dehydrogenation of formic acid as well as hydrogenation of nitro groups (Fig. s28). In the presence of formic acid and Pd NPs, the absorption peak of pNP disappeared at 400 nm, which indicated that the nitro group was reduced to an amino group (Fig. 5c). The reduction of nitro groups was also inhibited by $O_2$. In an $O_2$-saturated solution, the nitro group cannot be reduced within the first 15 min of the reaction but rapidly reduced in the following two minutes. With decreases in $O_2$ concentration, the plateau period decreased (Fig. 5d). This is because the initial potential of $O_2$ reduction on Pd NPs was much preferable to that of nitro groups; therefore, $O_2$ reduction proceeded first (Fig. 5e)[44]. When $O_2$ was depleted, nitro groups can serve as the electron acceptor and be reduced to amino groups. The phenomenon was similar to the previous reduction of FMN by formic acid in the presence of $O_2$ and Pt NPs.

## Discussion

In summary, we utilized ABTS$^{+\bullet}$ instead of $O_2$ as an electron acceptor and demonstrated that the oxidation of glucose proceeds via dehydrogenation on noble metal NPs. The reaction path of Au NPs catalysis is the same as that of natural GOD, except that OH$^-$ is used as a Brønsted base to abstract H$^+$ from glucose. The use of OH$^-$ as a Brønsted base corresponds to that glucose oxidation and alcohol oxidation are faster under alkaline conditions. Rotating disk tests confirmed that $O_2$ was preferable to be

reduced to $H_2O_2$ than $H_2O$ with Au NPs, and more likely to be reduced to $H_2O$ on the other noble metal NPs. These results explain the phenomenon that only Au NPs out of the six noble metal NPs can catalyze the oxidation of glucose and produce $H_2O_2$. In terms of the roles of the noble metal NPs, they catalyze both the dehydrogenation of hydroxyl groups and the ORR whose behavior is similar to that of FAD, thus we can define the noble metal NPs as coenzyme mimics. From the perspective of coenzyme mimics, noble metal NPs have been found to mimic various enzymatic electron transfer reactions for Cyt c and coenzymes as well as nitro compound reduction. It is also suggested that elucidating the catalytic process of natural enzymes and hunting for catalysts with similar catalytic functions are rational strategies to design more kinds of and higher efficient nanozymes.

## Methods

**Chemicals**. 3,3′5,5′-tetramethylbenzidine (TMB, 99%) was purchased from Sigma-Aldrich. Sodium borohydride (NaBH$_4$, 98%) was acquired from Aladdin. Glutathione (GSH, 98%) was bought from Genview. Glucose (99%) and gold chloride hydrate (HAuCl$_4$·4H$_2$O, Au ≥ 47.8%) were purchased from Beijing Chemical Works. All chemicals were used as received without further purification. Ultrapure water was used throughout the experiments.

**Characterization**. TEM images were obtained with a Hitachi H-8100 EM microscope operated at 100 kV. XPS measurements were performed on an ESCA-LABMKII (VG Co., UK) spectrometer with an Al Kα excitation source. UV–Vis absorption measurements were carried out on an Agilent Cary 60 (Varian) UV–Vis–near-infrared (NIR) spectrometer.

**Synthesis of noble metal NPs**. In a typical experiment, 0.05 mmol metal salt and 20 mg PVP were added to 45 mL water under vigorous stirring. After 5 min, 5 mL NaBH$_4$ solution (100 mM) was introduced. The reaction mixture was stirred at room temperature for 24 h. In particular, the Ir NPs were synthesized by reacting at 80 °C for 3 h and then stirring at room temperature for 24 h.

**Colorimetric detections of H$_2$O$_2$ produced by the noble metal NPs catalytic oxidation of glucose**. In a typical procedure, 50 μL of as-prepared noble metal NPs and 50 μL of glucose (1 M) were sequentially added into a vial containing 860 μL of 100 mM PBS buffer (pH = 7.4), After 20 min incubation at room temperature, 20 μL HRP (0.1 mg mL$^{-1}$) and 20 μL TMB (20 mM in DMSO:EtOH = 1:9) were

added into the above solution. UV–Vis absorption measurements were performed within 2 min.

**Detection of the oxidation products of alcohols**. In a typical procedure, 500 μL of as-prepared Au NPs and 20 μL of alcohol (benzyl alcohol or cyclohexanol) were sequentially added into a vial containing 10 mL of 100 mM PBS (pH = 8). After reaction for 2 h, 5 mL of ethyl acetate was added into the mixture solution to extract organic components. The products were detected by GC-MS.

**ABTS$^{+•}$ as the electron acceptor for glucose oxidation**. In a typical experiment, 1 mL of ABTS (10 mM in water) was mixed with 1 mL of $K_2S_2O_8$ (3.5 mM in water) and stored in dark for 12 h to obtain ABTS$^{+•}$. In total, 50 μL of as-prepared metal NPs, 50 μL of glucose (1 M) and 20 μL of ABTS$^{+•}$ solution were sequentially added into a vial containing 880 μL of 100 mM PBS (pH = 7.4). The reduction of ABTS$^{+•}$ was measured by the adsorption change at 734 nm.

**Catalase-like activity**. In a typical experiment, 50 μL of as-prepared noble metal NPs and 50 μL of $H_2O_2$ (1 M) were sequentially added into a vial containing 900 μL of 100 mM different buffer solutions (acetate buffer (pH = 5-6); PBS (pH = 7-8); carbonate buffer (pH = 9–10)). The decomposition of $H_2O_2$ was measured by the decrease of absorbance at 240 nm.

**EPR experiments**. In a typical experiment, 50 μL of as-prepared metal NPs, 50 μL of glucose (1 M) and 20 μL of DMPO solution were sequentially added into a vial containing 880 μL of 100 mM PBS (pH = 7.4). The mixture was deoxygenated by bubbling $N_2$ for 20 min before recording the EPR spectra.

**Electrochemical tests**. The ORR performance of the noble metal NPs was measured in 0.1 M KOH at room temperature. A rotating ring-disk electrode (RRDE) modified with noble metal NPs (980 μL of noble metal NPs mixed with 20 μL of 5 wt% Nafion® solution) on the disk served as the working electrode. Pt foil served as the counter electrode and Ag/AgCl electrode was used as the reference electrode. Before the electrochemical test, the electrolyte solutions were purged with $O_2$ (or $N_2$) for at least 30 min. The LSV plots were recorded by applying proper potential ranges at the scan rate of 10 mV/s. The electron transfer number ($n$) and selectivity of the noble metal NPs toward $H_2O_2$ formation can be calculated according to the well-known relation (Eqs. (1) and (2)):

$$n = 4 \frac{|I_{disk}|}{|I_{disk}| + I_{ring}/N} \tag{1}$$

$$H_2O_2\% = 200 \frac{I_{ring}}{N|I_{disk}| + I_{ring}} \tag{2}$$

**Colorimetric detection of Glucose by Au and PB**. In the experiment, 50 μL of as-prepared Au NPs and 100 μL of glucose in different concentrations were added into 850 μL of $Na_2CO_3$–$NaHCO_3$ solution (50 mM, pH = 10). The reaction mixture was incubated at room temperature for 1 h. Then 500 μL of the above solution, 20 μL of PB (5 mg mL$^{-1}$) and 20 μL ABTS (20 mM) were sequentially added into a vial containing 460 μL of $CH_3COOH$–$CH_3COONa$ buffer (100 mM, pH = 4). The reaction mixture was further incubated at room temperature for another 1 h before absorption spectroscopy measurement.

**Colorimetric detection of Glucose by ABTS$^{+•}$ reduction**. In a typical experiment, 50 μL of as-prepared Au or Pt NPs, 20 μL as prepared ABTS$^{+•}$ solution and 100 μL of glucose in different concentrations were added into 830 μL of PBS (50 mM, pH = 7.4). The reaction mixture was incubated at room temperature for 30 min, then tested by absorption spectroscopy.

**Hydride transfer from formate to FMN and Cyt $c$**. In a typical experiment, 50 μL of as-prepared noble metal NPs, 50 μL of HCOOK (5 M) and 20 μL of FMN (1 mM) or Cyt $c$ (50 μM) were sequentially added into a vial containing 880 μL of 20 mM HEPES buffer (pH = 7.4). The FMN reduction was monitored by the adsorption change at 450 nm. The Cyt $c$ reduction was monitored by the adsorption change at 550 nm.

**Catalytic reduction of $p$-NP by formate**. In a typical experiment, 50 μL of as-prepared noble metal NPs, 50 μL of HCOOK (5 M), and 20 μL of $p$-NP (1 mM) were sequentially added into a vial containing 880 μL of 20 mM HEPES buffer (pH = 7.4). The $p$-NP reduction was monitored by the absorption change at 400 nm.

**Oxygen-consumption assays**. In a typical experiment, 2 mL of as-prepared noble metal NPs, 4 mL of HCOOK (5 M) were sequentially added into a vial containing 34 mL of 20 mM HEPES buffer (pH = 7.4). $O_2$ concentration was measured by using a Clark-type oxygen electrode (Hansatech Instruments).

## Data availability

The data that support the findings of this study are available from the corresponding author upon reasonable request.

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

## Acknowledgements

This work was financially supported by the Ministry of Science and Technology of China (Nos. 2016YFA0203203, 2019YFA0709202), the National Natural Science Foundation of China (No. 22074137).

## Author contributions

J.C., Y.F., and S.D. conceived this project and designed the studies; J.C. prepared the samples and performed the catalytic tests; Q.M. and L.H. performed the electron microscopy experiments and data analysis; M.L., D.C., and W.W. helped with the data analysis. All authors discussed the experimental procedures and results. J.C., Y.F., and S.D. wrote the paper with contributions from all authors.

## Competing interests

The authors declare no competing interests.
