## [Peer Review File · Nature Communications]

Reviewers' Comments:

Reviewer #1:

Remarks to the Author:

The authors have investigated the mechanism for the catalytic reactions between glucose and O₂ (ABTS⁺), with noble metal nanoparticles (NPs) as the catalysts, which they call nanozymes. They have found that the most probable mechanism for this catalytic process is first the reduction of the NPs by the glucose to form the intermediate like NP-H and then is the transfer of the H from NP-H to the oxidant O₂ (ABTS⁺). I feel this work could be published after the authors address my following concerns.

- 1) Because the proposed catalytic mechanism involves mainly the transfer of hydrogen atoms rather than electrons and because electron transfer is a concept that already exists, the title of this manuscript is misleading.
- 2) The mechanism proposed by the authors for the catalytic process (glucose + O₂) is not complete. It could not explain how H₂O₂ is formed and why Au has the largest catalytic activity with the activity order Au>Pt>Ru>Ir>Pd>Rh.
- 3) It seems that the oxidations of TMB in presence of noble metals NPs and O₂ follow a totally different mechanism, because this series of reactions have a totally different activity order (Pt>Ir>Pd>Rh>Ru>Au vs. Au>Pt>Ru>Ir>Pd>Rh). The authors may want to explain the reason for the different activity order when the reactant is changed from glucose to TMB.

Minor points

1. Line 56 in Page 2, the abbreviation "GOH" is not defined before use.
2. Line 196, 197 and 199 in Page 6, what does "C1" and "N5" mean? Please label them in Figure 2d.

Reviewer #2:

Remarks to the Author:

In the manuscript titled "Open a way to explore nanozymes by following the electron transport pathway in the catalytic process of natural enzymes", the authors studied the glucose oxidase mimics of Au and other noble metal NPs. They found that all the NPs can catalyze and dehydrogenation of glucose, and only Au ones reduce O₂ to H₂O₂ instead of H₂O by two electrons transfer. Overall, this work is interesting and the experiments are rather comprehensive. So, it can be accepted for the publication on this Journal after concerning the following comments.

- (1) The authors have considered that glucose and other substrates (Figure 1g) were dehydrogenated and formed aldehyde groups in the first step reaction. It seems reasonable. However, in my opinion, to solid their contribution, the authors should design and present some experiment results to demonstrate the generation of aldehyde products.
- (2) The authors shown that their proposed catalytic systems can be employed for glucose detection by a colorimetric signal readout. In view of "practical" applications, for example blood sugar assay, two points should be well concerned. The first one is the catalytic performances of Au and other metal NPs glucose oxidase mimics. Based on previous reports (for example JACS, 2016, 138, 16645), the catalytic activity of Au NPs can be substantially quenched by thiol molecules due to strong S-Au bond. It is known many biological samples contain lots of thiol biomolecules (cysteine, glutathione, etc.). So, whether can these metal NPs well hold their activity in some complex environments? Second, as the observation by the authors, lots of hydroxyl-containing substances (Figure 1g) can be catalyzed and present similar reactions. Obviously, how to resolve the selectivity?

Reviewer #3:

Remarks to the Author:

This work by Chen et al. systematically demonstrated the glucose oxidase-like catalytic mechanism of noble metal nanozymes from the electron transfer pathway. They found that the glucose oxidation process catalyzed by Au NPs is the same as that of natural glucose oxidase, that is, a

two-step reaction, including glucose dehydrogenation and subsequent two electrons to reduce O₂ to H₂O₂. Moreover, Pt, Pd, Ru, Rh and Ir NPs can also catalyze the dehydrogenation of glucose, but O₂ tends to be reduced to H₂O. Using the electron transfer properties of noble metal nanoparticles, they overcome the limitation of the traditional two-step glucose analysis that must generate H₂O₂, and achieve rapid one-step colorimetric detection of glucose. Inspired by the electron transport pathway in the process of natural enzyme catalysis, it was also found that noble metal nanoparticles can mimic various enzymatic electron transfer reactions of cytochrome c, coenzyme and nitrobenzene reduction. This work explained the similarity of the catalytic mechanism between nanozymes and natural enzymes through simple and ingenious design, and further laid the foundation for the research of the enzyme-like catalytic mechanism of noble metal nanozymes. Although there have been many studies on the catalytic mechanism of nanozymes, this study is one of the rare studies that confirms the similarity between nanozymes and natural enzymes from the electron transfer pathway. This research will have important reference significance for subsequent research on the catalytic mechanism of nanozymes. Through an in-depth understanding of the electron transfer path in the catalysis of noble metal nanozymes, this research subtly improved the detection method of glucose and extended the application scope to various enzymatic electron transfer reactions. This research will promote the research and application of nanozymes in the field of molecular detection. Therefore, I suggest that this manuscript be considered for publication by Nature Communications after the following revisions.

1. The molecular formulas of hydroxyl radicals and superoxide radicals were not standardly written.
2. Some abbreviations in the text lack full explanations, such as GOH, NAD(P), etc.
3. In line 146, the authors defined Au NPs as formate oxidase mimics and alcohol oxidase mimics. However, Figure 1g showed that the oxidation rate of ethanol catalyzed by Au NPs was very low, so it is not appropriate to call it an alcohol oxidase mimic.
4. In line 179, it is mentioned that Au NPs did not exhibit catalase-like activity. However, many previous studies have shown that Au NPs have considerable catalase-like activity (Small, 2017, 13(26): 1700278; Small, 2016, 12(30): 4127-4135; Biomaterials, 2013, 34(3): 765-773.). How to explain the contradiction between this study and previous studies?
5. The Method section lacks specific experimental methods for detecting catalase activity.

Reviewer #1 (Remarks to the Author):

The authors have investigated the mechanism for the catalytic reactions between glucose and O₂ (ABTS⁺), with noble metal nanoparticles (NPs) as the catalysts, which they call nanozymes. They have found that the most probable mechanism for this catalytic process is first the reduction of the NPs by the glucose to form the intermediate like NP-H and then is the transfer of the H from NP-H to the oxidant O₂ (ABTS⁺). I feel this work could be published after the authors address my following concerns.

Response. Thank you very much for your positive comments.

Comment 1. Because the proposed catalytic mechanism involves mainly the transfer of hydrogen atoms rather than electrons and because electron transfer is a concept that already exists, the title of this manuscript is misleading.

Response 1. Thank you for this suggestion. The title of this manuscript has been changed to “Bionic catalytic pathway enlightened noble metal nanozymes”

Comment 2. The mechanism proposed by the authors for the catalytic process (glucose + O₂) is not complete. It could not explain how H₂O₂ is formed and why Au has the largest catalytic activity with the activity order Au > Pt > Ru > Ir > Pd > Rh.

Response 2. Thank you for this suggestion. The specific catalytic process is shown in Figure s19 (* denotes a bare surface active site). Glucose oxidation can be divided into two steps: the first step is dehydrogenation of glucose, and the second step is oxygen reduction. For dehydrogenation of glucose, the O-H bond in the hydroxyl group breaks and the hydrogen protons are removed. Then the C-H bond breaks and the proton and two electrons leave. The removed protons and electrons were transferred to oxygen. In the oxygen reduction reaction, O₂ can be reduced to H₂O by 4e⁻ pathways or to H₂O₂ by 2e⁻ pathways. The difference between the two paths is that the O = O double bond in the 2e⁻ path is not completely broken, so the product is H₂O₂, while the O = O double bond in the 4e⁻ path is completely broken, resulting in water.

For Au catalyzed oxygen reduction, the energy barrier of O = O double bond breaking is higher (*Chem. Rev.* 2018, 118, 5, 2816–2862). Therefore, Au catalyzed oxygen reduction mainly takes the 2e⁻ path, and the product is H₂O₂.

In previous reports, theoretical calculation shows that the rate determining step of alcohol oxidation is the breaking of C-H bond, and Au has the strongest effect on reducing the breaking energy of C-H bond (*Science* 330 (6000),74-78). Therefore, Au has the highest catalytic activity.

We have added some content in the revised manuscript to discuss the specific catalytic process of glucose oxidation.

This figure was added as Supplementary Figure 21 in the revised Supplementary Information.

Supplementary Figure 21. (a) Mechanism of glucose oxidation catalyzed by noble metal NPs. (b) The difference between 2e- path and 4e- path in oxygen reduction reaction.

Comment 3. It seems that the oxidations of TMB in presence of noble metals NPs and O_2 follow a totally different mechanism, because this series of reactions have a totally different activity order (Pt Ir Pd Rh Ru Au vs. Au Pt Ru Ir Pd Rh). The authors may want to explain the reason for the different activity order when the reactant is changed from glucose to TMB.

Response 3. The oxidations of TMB in presence of noble metals NPs and O_2 follow totally different mechanism. The oxidation potential of TMB is 1.13 V, which is only slightly lower than that of oxygen (1.23 V). Therefore, it is generally believed that the oxidation of TMB requires the production of reactive oxygen species with higher activity.

In previous studies, researchers reported that the oxidase-like activity of a metal is closely related to the dissociative adsorption of the O_2 on the respective metal surface (*Nat Commun* **9**, 129 (2018)):

According to this mechanism, the metal surface catalyzes the dissociation of the O_2 molecule to yield single O atoms, which are responsible for the subsequent oxidation of substrates.

Because Au is difficult to catalyze the break of double bonds in O_2 , it is difficult for Au to catalyze TMB oxidation with O_2 as oxidant.

In the revised manuscript, some discussion was added to speculate the reason of differences between glucose oxidation and TMB oxidation.

Minor points

- Line 56 in Page 2, the abbreviation “GOH” is not defined before use.

Response. Thank you very much for pointing this out. The “GDH” was defined before use in the revised manuscript.

2. Line 196, 197 and 199 in Page 6, what does “C1” and ”N5” mean? Please label them in Figure 2d.

Response. Thank you for your suggestion. We have noted “C1” and ”N5” in Figure 2d.

Figure 2d. Mechanism of glucose oxidation catalyzed by GOD.

Reviewer #2 (Remarks to the Author):

In the manuscript titled “Open a way to explore nanozymes by following the electron transport pathway in the catalytic process of natural enzymes”, the authors studied the glucose oxidase mimics of Au and other noble metal NPs. They found that all the NPs can catalyze and dehydrogenation of glucose, and only Au ones reduce O_2 to H_2O_2 instead of H_2O by two electrons transfer. Overall, this work is interesting and the experiments are rather comprehensive. So, it can be accepted for the publication on this Journal after concerning the following comments.

Response. We are very grateful to your valuable comments.

Comment 1. The authors have considered that glucose and other substrates (Figure 1g) were dehydrogenized and formed aldehyde groups in the first step reaction. It seems reasonable. However, in my opinion, to solid their contribution, the authors should design and present some experiment results to demonstrate the generation of aldehyde products.

Response 1. Thank you for your suggestion. We chose benzyl alcohol and cyclohexanol as substrates and detected the products by GC-MS. In the typical procedure, 500 μ L of as-prepared Au NPs and 20 μ L of benzyl alcohol or cyclohexanol were sequentially added into a vial containing 10 mL of 100 mM PBS (pH = 8). After reaction for 2 hours, 5 mL of ethyl acetate was added into the mixture solution to extract organic components. The products were detected by GC-MS.

As shown in Figure 1, benzaldehyde can be detected in the presence of Au, which proves that Au

can catalyzed the oxidation of alcohol hydroxyl to aldehyde group. Besides, when the substrate is cyclohexanol, the oxidation product is cyclohexanone.

Supplementary Figure 16. Chromatogram of benzyl alcohol before (a) and after (b) Au catalyzed oxidation reaction.

Supplementary Figure 17. Chromatogram of cyclohexanol before (a) and after (b) Au catalyzed oxidation reaction.

Comment 2. The authors shown that their proposed catalytic systems can be employed for glucose detection by a colorimetric signal readout. In view of “practical” applications, for example blood sugar assay, two points should be well concerned. The first one is the catalytic performances of Au and other metal NPs glucose oxidase mimics. Based on previous reports (for example JACS, 2016, 138, 16645), the catalytic activity of Au NPs can be substantially quenched by thiol molecules due to strong S-Au bond. It is known many biological samples contain lots of thiol biomolecules (cysteine, glutathione, etc.). So, whether can these metal NPs well hold their activity in some complex environments? Second, as the observation by the authors, lots of hydroxyl-containing substances (Figure 1g) can be catalyzed and present similar reactions. Obviously, how to resolve the selectivity?

Response 2. Thank you for your insightful comments. In our early study, we also found that Au is easy to inactivate. At the beginning of this work, we synthesized citric acid protected gold nanoparticles (Cit-Au). The size of Cit-Au is about 13 nm, but its catalytic activity is higher than that of PVP-Au (5 nm) in pure water. We think that the binding force of citric acid on Au surface is weak, therefore, Cit-Au has higher catalytic activity than PVP-Au. This phenomenon is consist with previous report that the catalytic activity of “Naked” Gold particles was very high (Angew. Chem.

Int. Ed. 2004, 43, 5812–5815). However, the Cit-Au are easy to inactivate, even in PBS. Although the activity of PVP-Au is low, it is more stable than Cit-Au. So we choose PVP-Au as catalyst to carry out the following research.

We further compare the stability of Au NPs under complex conditions. When 20 μL of serum was added to the reaction solution, the catalytic activity of Cit-Au disappeared completely, while that of PVP-Au remained about 10%. PVP has a good protective effect on nanoparticles, so PVP-Au can still retain a certain catalytic ability. According to our experiences, it is believe that even the same kind of nanoparticles may show different stability and catalytic ability (we can even think they are different catalysts, as you may see, different literatures with same kind of nanoparticles may give contradictive results). Our work tried to balance the activity and stability of the nanoparticles because PVP capped Au NPs are not easily quenched.

Improving the selectivity of substrate is always the challenge task for the development of pure nanozymes. Taking Au NPs for example, many works reported the applications of Au nanozymes for the detection of glucose, while these work hardly achieve good selectivity because nanozyme are essentially nanoparticles that may not provide a specific active site to bind the substrate. To fulfill the selectivity of the nanozyme, we can usually use them with further modifications, or innovatively use them in specific cascade reactions.

In this work, we proposed the protocol of synthesizing 6 kinds of metal nanozymes by a simple and cheap method, followed by exploring their catalytic mechanism, studying their catalytic ability and developing the cascade reactions for nanozyme catalysis. We also think the stability and selectivity of Au nanoparticles are important for bioassay, which may need systematic study to address. Moreover, we believe that revealing the mechanism of glucose oxidation catalyzed by Au NPs is of great significance for guiding studies to improve the catalytic selectivity in the future.

Reviewer #3 (Remarks to the Author):

This work by Chen et al. systematically demonstrated the glucose oxidase-like catalytic mechanism of noble metal nanozymes from the electron transfer pathway. They found that the glucose oxidation process catalyzed by Au NPs is the same as that of natural glucose oxidase, that is, a two-step reaction, including glucose dehydrogenation and subsequent two electrons to reduce O_2 to H_2O_2 . Moreover, Pt, Pd, Ru, Rh and Ir NPs can also catalyze the dehydrogenation of glucose, but O_2 tends to be reduced to H_2O . Using the electron transfer properties of noble metal nanoparticles, they overcome the limitation of the traditional two-step glucose analysis that must generate H_2O_2 , and achieve rapid one-step colorimetric detection of glucose. Inspired by the electron transport pathway in the process of natural enzyme catalysis, it was also found that noble metal nanoparticles can mimic various enzymatic electron transfer reactions of cytochrome c, coenzyme and nitrobenzene reduction. This work explained the similarity of the catalytic mechanism between nanozymes and natural enzymes through simple and ingenious design, and further laid the foundation for the research of the enzyme-like catalytic mechanism of noble metal nanozymes. Although there have been many studies on the catalytic mechanism of nanozymes, this study is one of the rare studies that confirms the similarity between nanozymes and natural enzymes from the electron transfer pathway. This research will have important reference significance for subsequent research on the catalytic mechanism of nanozymes. Through an in-depth understanding of the electron transfer path

in the catalysis of noble metal nanozymes, this research subtly improved the detection method of glucose and extended the application scope to various enzymatic electron transfer reactions. This research will promote the research and application of nanozymes in the field of molecular detection. Therefore, I suggest that this manuscript be considered for publication by Nature Communications after the following revisions.

Response. We are very grateful to your encouraging comments.

Comment 1. The molecular formulas of hydroxyl radicals and superoxide radicals were not standardly written.

Response 1. Thank you very much for the comments. These errors have been corrected.

Comment 2. Some abbreviations in the text lack full explanations, such as GOH, NAD(P), etc.

Response 2. Thank you very much for pointing this out. The “GDH” and “NAD(P)” were defined before use in the revised manuscript.

Comment 3. In line 146, the authors defined Au NPs as formate oxidase mimics and alcohol oxidase mimics. However, Figure 1g showed that the oxidation rate of ethanol catalyzed by Au NPs was very low, so it is not appropriate to call it an alcohol oxidase mimic.

Response 3. According to the advice, the language about alcohol oxidase mimic have been removed from the manuscript.

Comment 4. In line 179, it is mentioned that Au NPs did not exhibit catalase-like activity. However, many previous studies have shown that Au NPs have considerable catalase-like activity (Small, 2017, 13(26): 1700278; Small, 2016, 12(30): 4127-4135; Biomaterials, 2013, 34(3): 765-773.). How to explain the contradiction between this study and previous studies?

Response 4. Thank you for your insightful question. We have read these papers carefully, and thought over the point you mentioned.

First, the catalytic performance of nanoparticles prepared by different synthesis methods will be different. Changing the synthesis methods is a common way to improve the activity or selectivity of catalysts.

Second, at present, there is no standard to measure the catalytic activities of nanozymes. Due to the lack of comparison with representative materials, it is difficult to accurately measure the catalytic activities of Au NPs in these articles. For a catalyst with poor activity, the catalytic results can be trickily obtained by increasing the amount of catalyst and prolonging the reaction time.

Third, a previous report (Biomaterials 48 (2015) 37-44, DOI:10.1016/j.biomaterials.2015.01.012) confirms our results and the two points mentioned above. The researcher compared the catalytic activities of Pt, Pd, Ag and Au for the decomposition of H₂O₂, and found that Au and Ag had no detectable catalytic effect compared with Pt and Pd.

Therefore, we conclude that Au is not a good catalyst for the decomposition of H₂O₂ in our work.

Figure response. The catalase-like activities of Pt, Pd, Ag, and Au at pH = 7.4. (from Biomaterials 48 (2015) 37-44).

Comment 5. The Method section lacks specific experimental methods for detecting catalase activity.

Response 5. For detecting catalase-like activity, 50 μL of as-prepared noble metal NPs and 50 μL of H_2O_2 (1 M) were sequentially added into a vial containing 900 μL of 100 mM buffer solution: acetate buffer (pH = 5-6), PBS (pH = 7-8), carbonate buffer (pH = 9-10). The decomposition of H_2O_2 was measured by the decrease of absorbance at 240 nm.

We have added this information to the revised Method section.

Reviewers' Comments:

Reviewer #1:

Remarks to the Author:

This work has for the first time disclosed the mechanism how molecular oxygen is catalytically activated by gold nanoparticles, which has confused me for several years. The authors have satisfactorily responded all my concerns. I feel the revised manuscript could now be accepted for publication at Nature Communication.

Reviewer #2:

Remarks to the Author:

In the revised version, the authors well modified their manuscript, which substantially enhanced its quality. My remaining only concern is the part of "Glucose detection" (page 7). Considering that the catalytic reaction of glucose oxidation is short of selectivity, the corresponding detection is probably "unpractical" for real samples. So, the authors should well deal with this part.

Reviewer #3:

Remarks to the Author:

The authors have addressed most concerns of the reviewers.

However, I think the authors should add the discussion/introduction about the shortcomings of Au nanozyme for practical applicaitons. These descriptions may help the readers better undersatnd why the rational design of nanozyme is in urgent demand.

In addition, the 'excellent glucose oxidase mimic'in abstract shoudl be revised, unless the authors can provide the data support that the enzymatic units between Au nanozyme and natural GOD are comparable.

After addressing these minor concerns, I think this work can be accepted by Nature Communications.

Reviewer #1 (Remarks to the Author):

This work has for the first time disclosed the mechanism how molecular oxygen is catalytically activated by gold nanoparticles, which has confused me for several years. The authors have satisfactorily responded all my concerns. I feel the revised manuscript could now be accepted for publication at Nature Communication.

Response: We appreciate your kind comments and thoughtful suggestions in improving the quality of our study.

Reviewer #2 (Remarks to the Author):

In the revised version, the authors well modified their manuscript, which substantially enhanced its quality. My remaining only concern is the part of “Glucose detection” (page 7). Considering that the catalytic reaction of glucose oxidation is short of selectivity, the corresponding detection is probably “unpractical” for real samples. So, the authors should well deal with this part.

Response: Thank you for your insightful comments. Achieving good selectivity to substrates has always been a difficult task in the field of nanozymes. Therefore, although there are many reports about the application of Au NPs in the detection of glucose, most literatures avoid mentioning selectivity.

On the contrary, we disclose that Au NPs can catalyze the oxidation of different kinds of sugars without obvious selectivity. This discovery is a drawback for Au NPs, whereas it is also a good omen to attract researchers to solve the bottleneck in practical detection of real samples.

Therefore, we stated the shortcomings of Au NPs in selectivity, and hope to attract the attention of relevant researchers. The discussion is added in the main text:

It should be mentioned that these demonstrated model detections of glucose are still far from real sample detections that require excellent selectivity and stability of NPs, which may be overcome by the further rational modifications of NPs as well as designing cascade reactions with NPs.

We appreciate your efforts in improving the manuscript.

Reviewer #3 (Remarks to the Author):

The authors have addressed most concerns of the reviewers.

However, I think the authors should add the discussion/introduction about the shortcomings of Au nanozyme for practical applications. These descriptions may help the readers better understand why the rational design of nanozyme is in urgent demand.

Response: Thank you for this suggestion. According to our experimental results, the low substrate selectivity hinders the application of Au nanozyme in practical glucose detection. To make it clear, further discussion about the shortcomings of Au nanozyme is added in the manuscript:

It should be mentioned that these demonstrated model detections of glucose are still far from real sample detections that require excellent selectivity and stability of NPs, which may be overcome by the further rational modifications of NPs as well as designing cascade reactions with NPs.

In addition, the 'excellent glucose oxidase mimic' in abstract should be revised, unless the authors can provide the data support that the enzymatic units between Au nanozyme and natural GOD are comparable.

Response: Thank you for pointing this out. The mass activity of Au nanozyme is lower than that of GOD. The 'excellent' have been removed from the manuscript.

After addressing these minor concerns, I think this work can be accepted by Nature Communications.

We thank you for your positive comments and recommendation.